# Cryopreservation of Sperm from an Endangered Snake with Tests of Post-Thaw Incubation in Caffeine

**DOI:** 10.3390/ani12141824

**Published:** 2022-07-17

**Authors:** Mark R. Sandfoss, Jessica Cantrell, Beth M. Roberts, Steve Reichling

**Affiliations:** Conservation and Research Department, Memphis Zoo, Memphis, TN 38112, USA; jcantrell@memphiszoo.org (J.C.); broberts@memphiszoo.org (B.M.R.); sreichling@memphiszoo.org (S.R.)

**Keywords:** assisted reproductive technologies, ex situ conservation, *Pituophis ruthveni*, biobank, captive breeding, bovine serum albumin, cryoprotective agent

## Abstract

**Simple Summary:**

Cryopreservation of sperm from reptiles to aid the recovery of endangered species continues to be a challenge. In this study, we tested the cryoperformance of a cryoprotective agent (CPA) mixture to cryopreserve sperm from the endangered Louisiana pinesnake (*Pituophis ruthveni*). The mixture contained Lake’s buffer with 10% *N*,*N*-dimethyl formamide (DMF), 2% methanol, 5% clarified egg yolk, (*v*/*v*% final concentration) and was tested against 16 experimental mixtures containing variable concentrations and mixtures of diluents, extenders, CPAs, and additives. In addition, we investigated the effects of post-thaw incubation on sperm motility in TL HEPES supplemented with 10% fetal bovine serum (H10) alone or supplemented with caffeine. We found that the majority of our test additives did not significantly improve the post-thaw motility or viability of sperm. The best performing experimental CPA mixture contained Lake’s buffer with 10% DMF, 2% methanol, and 5% clarified egg yolk with the addition of 5 mg/mL bovine serum albumin (BSA), and post-thaw incubation in both H10 and H10 with caffeine showed improved forward motility. Cryopreservation of sperm from the Louisiana pinesnake improved with the addition of BSA to our base CPA mixture, and post-thaw incubation in H10 improved with caffeine.

**Abstract:**

Cryopreservation of sperm to preserve the genetic diversity of declining populations is a promising technique to aid in the recovery of endangered species such as the Louisiana pinesnake (*Pituophis ruthveni*). However, this technique has been performed on only a handful of snake species and with limited success. Here, we tested a cryoprotective agent (CPA) mixture containing Lake’s buffer with 10% *N*,*N*-dimethyl formamide (DMF), 2% methanol, 5% clarified egg yolk, (*v*/*v*% final concentration) against 16 other CPA-treatment mixtures. These contained either Lake’s buffer or TEST egg yolk buffer as the base diluent with a penetrating or non-penetrating CPA on the post-thaw recovery of sperm motility and viability. We also investigated the effect of post-thaw incubation treatment in TL HEPES supplemented with 10% fetal bovine serum (H10) alone or with caffeine on post-thaw motility parameters. Sperm from 16 Louisiana pinesnakes was cryopreserved, and the effectiveness of the CPA treatment mixtures and post-thaw treatments was determined based on measurements of sperm motility and viability. Sperm cryopreservation significantly reduced initial post-thaw sperm quality for all of the extender treatments. Viability of sperm was best maintained when cryopreserved in an CPA treatment mixture containing Lake’s buffer with 10% DMF, 2% methanol, and 5% clarified egg yolk with the addition of 5 mg/mL bovine serum albumin (BSA). For several extender mixtures a similar percent of post-thaw motility was observed, but no forward motility returned in any post-thaw samples prior to incubation in dilution treatments. Following incubation in both post-thaw treatments, the percent of forward motility and the index of forward progressive movement improved significantly. Post-thaw dilution with H10 containing caffeine improved motility parameters over H10 alone, suggesting further investigation of post-thaw treatment in caffeine could be beneficial. Although, cryopreservation of sperm from the Louisiana pinesnake continues to present a challenge, post-thaw dilution and the addition of BSA to CPA mixtures provides areas for improving cryopreservation methods for this endangered species.

## 1. Introduction

Biodiversity is in decline across the globe [1], and conservation efforts continue to seek strategies to mitigate threats and recover threatened species [2]. Cryopreservation of sperm and the development of assisted reproductive technologies (ARTs) can be used to preserve the genetic diversity of declining populations [3,4,5,6,7] and aid in their recovery [8,9,10]. Successful examples of the use of ARTs in conservation programs can be found primarily in mammals [11] and birds [12,13], which is probably a direct result of their extensive development and use as livestock [14,15]. However, there is an urgent need to develop cryopreservation methods and associated ARTs across a wider range of species [16] because members of all major taxonomic groups are experiencing some type of decline.

As a taxonomic group, non-avian reptiles are incredibly speciose (>11,000 species) and many are threatened with extinction [17,18]. Despite this, there has been a relatively limited number of investigations into cryopreservation [7,19,20]. Published studies of the cryopreservation of reptile sperm have reported low levels of recovery for motility and viability post-thaw [21,22,23,24,25]. This may be due in part to the filiform shape of reptilian sperm [26], which might make them more vulnerable to cryo-injury [27]. However, several recent studies have produced encouraging results [28,29,30]. The diversity of reproductive systems found across reptilia suggests that cryopreservation might need to be tailored for specific taxonomic groups or mode of reproduction. Therefore, we need to conduct tests on a broad range of species to better understand if general principles exist and to identify common aspects of sperm cryopreservation methods in reptiles. For example, a recent comparative study of several species of snake found that the cryoprotective agent (CPA) mixture that provided the best recovery of post-thaw sperm motility varied by family [29], while an investigation of the toxicity of commonly used CPAs to freshly diluted sperm from two closely related species differed markedly in their responses [31]. Comparative studies that identify broadly applicable cryopreservation methods are indeed of significant scientific value, but the seemingly taxon-specific findings of recent studies suggest that initial efforts to develop cryopreservation methods should focus on species that are currently threatened with extinction. The Louisiana pinesnake, *Pituophis ruthveni*, is currently listed as threatened under the *Endangered Species Act* (2018; 83 FR 14958) and as endangered by the Convention on International Trade in Endangered Species (CITES) [32]. As such, an ex situ captive breeding program has been established at several zoological parks including the Fort Worth Zoo (Ft. Worth, TX, USA), where captive adult males provide an opportunity to test cryopreservation methods. The process often causes sublethal damages: reduction in sperm motility and acrosome integrity, decrease in mitochondrial membrane potential, increased reactive oxygen species, lipid peroxidation, and damage to DNA [33]. The addition of CPAs is meant to mitigate these effects during the freezing and thawing process [34]; however, identifying the most effective CPA mixture can be difficult due to the wide range of metabolic and biophysical effects that, although necessary for their modes of action, can further complicate cell function. Furthermore, sperm are highly specialized cells that show an incredible diversity in morphology [35] and varied responses to cryopreservation [36]. Initial tests to explore the toxic cryoprotectant effects of six commonly used membrane-permeable CPAs at three concentrations found that the motility of fresh Louisiana pinesnake sperm samples were highly sensitive although the concentration and time of effect varied by CPA type [31]. In freezing tests of the same six CPAs added to the semen diluent, the CPA type and concentration that performed the best had post-thaw motility recovery as high as 25 versus 0% for sperm frozen with semen diluent alone. Interestingly, the CPA and concentration combinations that were the least toxic were not necessarily the most cryoprotectant [31]. Further testing of permeable and non-permeable CPAs, diluents, and additives is needed to investigate alternative methods to improve cryosurvival and recovery of motility in this species.

The cryopreservation of sperm imposes numerous stresses beyond physical damage. Energy production to support motility is important for successful cryopreservation. Sperm cells obtain their energy through two main pathways: oxidative phosphorylation and glycolysis. The sperm of many species are predicted to be able to switch pathways depending on conditions in the female reproductive tract, substrate, and oxygen concentrations [37,38]. Sugars, which were perhaps the first additives to be used in cryopreservation [39,40], play various roles during freezing and thawing of sperm, including as an energy source and as a cryoprotectant [41]. Trehalose and fructose have been effective as non-permeating CPAs for cryopreservation in several species [41,42]. Additionally, investigations into cryopreservation using CPA treatment mixtures containing the alternative energy substrates sodium lactate and sodium pyruvate showed positive benefits for cattle [43] and boars [44]). Sodium pyruvate, which is thought to provide energy, may also act as an antioxidant to scavenge hydrogen peroxide, produced during lipid peroxidation, to reduce cell damage [45].

Antioxidants are an important feature of modern CPAs because the cold shock during the freezing–thawing process increases oxidative stress induced by free radicals [46,47]. Amino acids found in seminal plasma have various functions including reducing free radicals, protecting cells against denaturation, and providing an oxidizable substrate [48]. However, the identities and roles of these amino acids during cryopreservation are not fully understood. Several, including glycine [49], have been tested as CPAs for various species with positive results [50,51,52]. Antioxidants derived from biological fluids such as bovine serum albumin and fetal bovine serum have also been shown to benefit sperm survival during cryopreservation [53,54].

Additionally, the effectiveness of post-thaw additives to improve the recovery of sperm motility in snakes remains untested. A recent investigation of cryopreservation in a varanid lizard found that the recovery of post-thaw motility improved following dilution of post-thaw semen into Dulbecco’s phosphate-buffered saline (PBS; Ca^++^, Mg^++^ free) (Sigma-Aldrich, St. Louis, MO, USA) containing caffeine [30], which had a similarly positive effect on mammalian sperm motility [55].

For this study, we tested the cryoprotective effects of 16 CPA treatment mixtures consisting of a base diluent (Lake’s solution or TEST egg yolk buffer) with varying concentrations of permeating and non-permeating CPAs plus additives thought to provide energy and antioxidant effects. At thaw, we investigated improved post-thaw motility following a fresh buffer dilution with and without caffeine. Our objectives were, first, to test the cryoprotective performance of a variety of mixtures comprised of different types and concentrations of CPAs (with and without additional energy and antioxidant additives) and, second, to test the effect of post-thaw incubation in a buffer with and without caffeine on the recovery of motility. We hypothesized that the CPA type and concentration would affect cryosurvival and post-thaw sperm motility without a clear prediction for the best treatment. Based on results for lizards and some species of mammals, we predicted that post-thaw incubation in a buffer with caffeine would improve motility. The aim of this research was to improve the cryopreservation of an endangered species and provide additional insight into possible methods for snakes, specifically, and reptiles generally.

## 2. Methods

### 2.1. Animals and Location

The Louisiana pinesnake is a large-bodied, non-venomous colubrid endemic to longleaf pine habitats in Louisiana and Texas. Semen was collected from 18 adults ranging in age from 3 to 20 years and weighing 945–3180 g. The snakes were located at the Fort Worth Zoo (Ft. Worth, TX, USA), where they were housed separately in large plastic cages, and fed weekly with commercially produced chicks, rats or a combination of the two based on the snake’s size and feeding behavior. Water was provided ad libitum. The photoperiod and temperature were adjusted throughout the year to simulate natural conditions within the native range of the species and to encourage the reproductive cycle. Semen collection occurred following the end of the brumation period, 1–4 March 2021. All research was approved by representatives of the Institutional Animal Care and Use Committee of both participating zoos (Memphis Zoo IACUC #2020-4, FWZ IACUC #2021-01-19).

### 2.2. Semen Collection Methods

Semen was collected using a modified ventral massage technique [56]. To encourage defecation prior to expressing semen, individual snakes were placed in a bucket with a small amount of lukewarm water. For semen collection, they were removed from the bucket, dried, and the upper portion of the body was hand-restrained while a second person applied pressure to the ventral side of the snake starting at the mid-body and slowly moving towards the cloaca. Pressure was then localized anterior to the cloaca until semen was expelled from the vas deferens into the open cloaca, where it was collected by a micropipette and transferred to tubes containing semen diluent H10, consisting of TL HEPES (Caisson Laboratories Inc. #IVL01, Caisson Laboratories, Smithfield, UT, USA) supplemented with 10% Fetal Bovine Serum (FBS, *v*/*v*, Sigma-Aldrich #F7524). There was no need to evert the hemipenes. Sample color, consistency, and volume were recorded as collected. Multiple fractions were collected per male with pipette tips and tubes being exchanged as the fluid expelled changed color and consistency. The time from the start of massage to ejaculation varied; however, a 20 min limit for hand restraint and ventral massage was maintained. We collected from four to five males a day. Ejaculates from individual snakes were stored on ice at 4 °C from time of collection until analysis and freezing (≤4 h).

### 2.3. Semen Analyses

Semen samples from each snake were examined using phase-contrast light microscopy. Fractions with similar consistency, motility, and concentration were combined per male and then further diluted 1:8 to 1:10 (semen: total volume) with H10 depending on sample volume and sperm concentration. The samples were assessed for percent sperm motility, forward progressive motility (FPM), and percent viability.

Percent motility (moving/moving forward/non-moving) was calculated as the number of moving sperm out of 100 randomly observed sperm at 400× magnification. The number of forward-moving motile sperm was simultaneously recorded and an index score for FPM was assigned on a scale of 0 to 5 (0 = no-motility; 5 = fast and straight-line motility). The percent of viabile sperm was determined using an eosin–nigrosin-based live–dead stain to measure membrane integrity (Jorvet Stain, Jorgensen Laboratories, Inc., Loveland, CO, USA) and counting the first 100 sperm at 400× magnification. All assessments of were completed using either an Olympus BX60 or CX41(Olympus Corp., Tokyo, Japan) phase contrast microscope. The concentration of sperm in the samples for each male was determined using fixed sperm (1% paraformaldehyde/saline) counted by a hemocytometer (Bright-Line, American Optical Corp., Buffalo, NY, USA).

### 2.4. Cryopreservation of Sperm

We tested the cryoprotective capabilities of 17 CPA treatment mixtures (Table 1) to maintain post-thaw viability and sperm motility. The test CPA treatment mixture comprised a base diluent (Lake’s solution [57] or TEST yolk buffer [58] (TEST Yolk refrigeration media, Irvine Scientific, #90129)); one or a combination of up to four membrane-permeating cryoprotectants (*N*,*N*-dimethylacetamide (DMA; Sigma-Aldrich #270555), *N*,*N*-dimethyl formamide (DMF; Sigma-Aldrich # 227056, Sigma-Aldrich, St. Louis, MO, USA); methanol (Fisher Scientific #A412), glycerol (Fisher Scientific #G33)); two non-membrane-permeating cryoprotectants (trehalose (Sigma-Aldrich #T9449) and fructose (Sigma-Aldrich #F3510)); and multiple additives: the amino acid glycine (Sigma-Aldrich #410225), bovine serum albumin (BSA; Sigma-Aldrich #9048468), fetal bovine serum (FBS; Sigma-Aldrich #F7524), Na-lactate (Sigma-Aldrich #L7022), and Na-pyruvate (Sigma-Aldrich #P2256). Additionally, egg yolk, which is the most widely used cryoprotectant for sperm, was included in all CPA treatment mixtures.

Due to the relatively small volumes of semen collected (<0.1 mL), the snakes were randomly divided into four experimental groups (Table 1) and each group had semen frozen in four unique CPA treatment mixtures and a CPA mixture (Table 1) held constant across all experimental groups to ensure comparable freezing performance (Appendix A) and referred to as the “CPA control” mixture. The CPA control was based on toxicity tests [31] and previous performance during pilot studies of cryopreservation in this species (MRS unpublished data). We then explored how this CPA control base mixture could be improved by the addition of energy sources, antioxidants, and non-membrane permeating CPAs and by varying the concentration and combination of permeating CPAs. These experimental tests were comprised of three categories (Table 1): (1) the addition of non-permeable CPAs, amino acids, or energy source to the CPA control base (CPA ID #3–8); (2) the addition of glycerol below toxicity levels to the CPA control base (CPA ID #9–12) or the addition of glycerol at a cryoprotectant concentration (CPA ID #13–17) [25,29,31] in both Lake’s and TEST egg yolk buffers; and (3) a comparative test of the CPA control mix of 10% DMF + 2% methanol using TEST egg yolk buffer (CPA ID #2, Table 1).

To test cryosurvival, the fresh semen, as collected and assessed as described above, was chilled to 4 °C, and then frozen in each of the CPA treatment mixture as follows: aliquots of the chilled semen diluted 1:8 to 1:10 (total volume with H10 depending on sample volume and sperm concentration) were extended 1:1 in each CPA treatment mixture via two successive additions of ½ CPA treatment volume to semen added 5 min apart at 4 °C to reduce osmotic shock [59]. After the 10 min mixing period, the semen-treatment mixtures were drawn into 0.25 mL cryostraws, sealed with critoseal putty (Fisher Scientific), and allowed to equilibrate for an additional 10 min at 4 °C. To freeze, the cryostraws were placed on a wire rack 10 cm above liquid nitrogen for 10 min and cooled to –80 °C at an average rate of approximately –8.7 °C/min [31]. The straws were then plunged directly into liquid nitrogen, loaded into canes, and stored in liquid nitrogen at –195 °C until thawing ($\bar{x}$ = 89 ± 40 days). Two replicate straws per male were frozen for each CPA treatment mixture and thawed on the same day for post-thaw assessments.

The straws (two per treatment per snake) were individually removed from the liquid nitrogen and thawed for 10 s at room temperature followed by submersion in a water bath at 40 °C for 10 s. Measures of motility and viability were repeated on the thawed sperm as described above.

### 2.5. Post-Thaw Treatments

The thawed sample for each straw, the “raw sample”, was analyzed and then divided into aliquots and diluted 1:1 (*v*/*v*) in H10 with or without 20 mM of caffeine (Sigma-Aldrich #C0750) to achieve a final caffeine concentration of 10 mM. Following a 30 min incubation at room temperature (23 °C), measures of motility were made as previously described, and the two treatment aliquots were compared to the raw sample to evaluate the effects of post-thaw dilution on the recovery of sperm motility.

### 2.6. Statistical Analyses

We used Kruskal–Wallis non-parametric tests to investigate the cryoprotection provided by each CPA treatment mixture for % total motility, % forward-moving motile sperm, FPM index, and % sperm viability. The effect of post-thaw dilution on sperm motility (% total motility, % forward-moving motile sperm, and FPM index) were also compared using Kruskal–Wallis non-parametric tests. All statistical tests were performed using program R (version 3.6.0) and statistical significance was set at a *p* < 0.05.

## 3. Results

### 3.1. Semen Collection

Semen was successfully collected from all 18 males, but two did not produce samples of sufficient quality for freezing (e.g., low concentration or sperm motility). Two males (“m4” and “m9”) produced a large enough volume of semen ($\bar{x}$ = 32 ± 14 μL) to be used in two CPA experimental groups (Table 1). Average total motility of fresh semen samples was 76 ± 10%. We found 87 ± 10% of motile sperm in fresh samples to be forward moving and the average FPM index assigned was 4.2 ± 0.5 (Table 2). We found no statistical differences in measures of motility for fresh semen between the experimental groups (Table 2). The average viability of sperm prior to freezing was 73 ± 17%, which was significantly different among the experimental groups (Table 2); therefore, the statistical analysis of viability was conducted using the measure of recovered viability calculated as the % viability of the post-thaw sample/% viability of fresh sample for each snake.

### 3.2. Cryopreservation

We found a significant effect of the type of CPA treatment mixture used on the recovery of viability (Kruskal–Wallis, *H* (16) = 51.35, *p* < 0.001) and the total motility of sperm post-thaw (Kruskal–Wallis, *H* (16) = 65.02, *p* < 0.001). No sperm were observed to be swimming in a forward direction for any raw post-thaw samples, so no statistical comparisons could be performed on the FPM index or % motile sperm. The highest average for total motility post-thaw was achieved using CPA treatment mixture #3 with 19.9 ± 6.0%, but several CPA treatment mixtures returned similar levels of total motility, and the CPA control returned 12.6 ± 7.2% (Figure 1A). The recovery of viability was the highest (51.7 ± 10.2%) when cryopreserved with CPA treatment mixture #3 (Figure 1B), which contained the diluent Lake’s buffer with the membrane-permeating CPA DMF at 10% and methanol at 2% plus 5% clarified egg yolk, and 5 mg/mL BSA. The CPA control recovered on average 27.2 ± 13.5% viability and several CPA treatment mixtures recovered very low (<3%) viability including #6 (2.2 ± 2.1%), #11 (1.1 ± 1.4%), #12 (1.7 ± 2.1%), #13 (2.5 ± 1.7%), and #14 (2.8 ± 3.9%) (Figure 1B).

### 3.3. Post-Thaw Additives

The effect of sperm dilution post-thaw in H10 with or without caffeine was not uniform across the measures of motility. Percent total motility was not significantly different following incubation in either post-thaw treatment (Kruskal–Wallis, *H* (2) = 0.32, *p* = 0.851) (Figure 2A). However, both the percent of motile sperm found to be moving forward (Kruskal–Wallis, *H* (2) = 18.2, *p* < 0.001),) and FPM index (Kruskal–Wallis, *H* (2) = 18.8, *p* < 0.001) increased significantly following incubation in H10 with and without caffeine (Figure 2B,C).

## 4. Discussion

This was the first study to investigate a wide range of semen extenders, CPAs, and post-thaw dilution techniques on semen collected from live males of an endangered species of snake. Cryopreservation in reptiles has been challenging, and we found recovery of viability and post-thaw motility of sperm from the Louisiana pinesnake to be the best when using the CPA control base mixture with the addition of BSA (CPA #3 in Table 1) (Lake’s buffer and the CPAs DMF at 10% and methanol at 2% with 5% clarified egg yolk plus 5 mg/mL BSA). The presence of BSA appeared to improve post-thaw motility and maintain viability during what is likely the first test of BSA for cryopreservation of sperm from a reptile. A primary function of BSA is the elimination of free radicals generated by oxidative stress [53]. In mammals, BSA has been found to benefit the motility, viability, and acrosome integrity of both fresh and frozen-thawed sperm [32,60,61].

Interestingly, no other additives to the base mix (CPAs #4–8) conferred additional cryoprotection for motility or viability For example, FBS––a complex solution containing growth factors, amino acids, sugars, lipids, vitamins, trace elements, and hormones––is commonly used in cell tissue preservation but did not provide any noticeable benefits during cryopreservation. FBS has been found to maintain high post-thaw progressive motility and to reduce production of free radicals in the semen of chickens [54] although because of the natural variation inherent in the composition of semen, we lacked precise information on its protective mechanisms [62]. However, freshly extracted semen that had been immediately diluted in H10 containing 10% FBS prior to a 1:1 dilution with the test extenders for freezing, might have reached a threshold of received benefits prior to freezing, which is why further increases of FBS in freezing extenders did not provide any additional cryoprotective benefits.

Similarly, the extender mixture of the diluent Lake’s buffer, 10% DMF, 2% methanol, and 5% egg with the addition of sodium lactate and sodium pyruvate (CPA #7), did not significantly improve cryosurvival. These mitochondrial substrates were thought to be an additional energy source for sperm of species that primarily use oxidative phosphorylation to maintain motility [63]. The lack of observed positive effects on sperm motility suggests that glycolysis may be more important for the Louisiana pinesnake during freezing and thawing; however, we cannot say definitively if the snake sperm primarily used glycolysis or oxidative phosphorylation to maintain motility during cryopreservation. We also observed that cryopreservation in the presence of the sugars trehalose and fructose did little to improve post-thaw motility or viability of sperm for this species, which contrasts with findings reported for species from other taxonomic groups [64,65].

Mixtures of CPAs that kept the same final CPA concentration, have been shown to be less toxic [66] and more cryoprotectant [25] relative to equal concentrations of each CPA on their own in other species; however, a previous investigation of the toxicity of mixed CPAs on fresh semen from Louisiana pinesnakes did not produce similar results [31]. Here, we directly tested the freezing performance of CPA mixtures with a lower concentration of DMF and with added glycerol (or glycerol and DMA) (CPAs #9–13). These CPA mixtures did not improve post-thaw motility with or without additional additives. In addition, our results suggest that extender mixtures with the highest number of additives or concentration of CPAs often resulted in very low post-thaw motility even if viability was maintained. And this was a common feature of the majority of CPA mixtures that recovered little to no post-thaw motility (CPAs #8, 11, 12, 13, 17).

We found that Lake’s buffer and TEST egg yolk buffer performed similarly as CPA diluents. However, the motility of raw samples post-thaw was generally low. TEST egg yolk buffer and PBS are the diluents most often cited for use in cryopreservation across all species of reptiles (e.g., [20,25,30]. We previously found poor results with PBS (MRS unpublished data) and, as seen here, TEST yolk performed similarly to Lake’s buffer as a diluent for the Louisiana pinesnake. In the only other known test of Lake’s solution as a diluent for cryopreservation in snakes, Zacarotti et al. [23], found no post-thaw motility; however, Lake’s was used in combination with the poorly performing CPA dimethyl sulfoxide (DMSO) at a low concentration (2 and 4%) [31]. We found that the extender mixture containing TEST yolk plus glycerol [29,67] conferred a similar recovery of viability as the CPA control mixture containing Lake’s buffer with 10% DMF, 2% methanol, and 5% egg. We are not aware of any cryopreservation study that has used this combination of extender and CPA mixture to cryopreserve sperm of a species of snake other than the Louisiana pinesnake.

Post-thaw incubation in H10 with and without caffeine resulted in very little improvement in total motility, but both post-thaw treatments increased the percentage of sperm moving forward. Some caution is warranted, however, as the observed effect on forward motility was statistically significant, but in absolute terms it was limited to changes of +1.5% following incubation in H10 alone and +2% in H10 with caffeine relative to observed forward motility of 0% for raw samples. Post-thaw dilution did significantly increase the FPM index, and H10 with caffeine was particularly effective in improving the speed of sperm as incubation in caffeine more than doubled the speed of the FPM relative to H10 alone. In mammals, forward motility is an indicator of fertility [68], and the use of caffeine to improve motility of cryopreserved sperm might benefit future attempts at artificial insemination for this species.

## 5. Conclusions

In this study, we found that a 12% final concentration of CPA, which included a mix of two CPAs (10% DMF plus 2% methanol), and BSA performed the best overall. Previously published studies of cryopreservation in snakes identified glycerol, DMF, and DMSO as possible cryoprotectants [22,24,25,29,31]. These studies all found support for a final concentration of CPA between 8 and 20%. The recovery of motility and viability in post-thaw samples was low in this study, and investigations into cryopreservation in reptiles should continue to test CPA mixtures and post-thaw incubation methods to improve recovery and identify patterns of success.

Biobanks or “animal arks” can preserve the genetics and viable reproductive tissues of an impressive amount of biodiversity with a much lower space requirement than a zoo. Cryopreservation and associated ARTs have incredible potential as conservation tools and need to be developed for a wider range of species.

## Figures and Tables

**Figure 1 animals-12-01824-f001:**
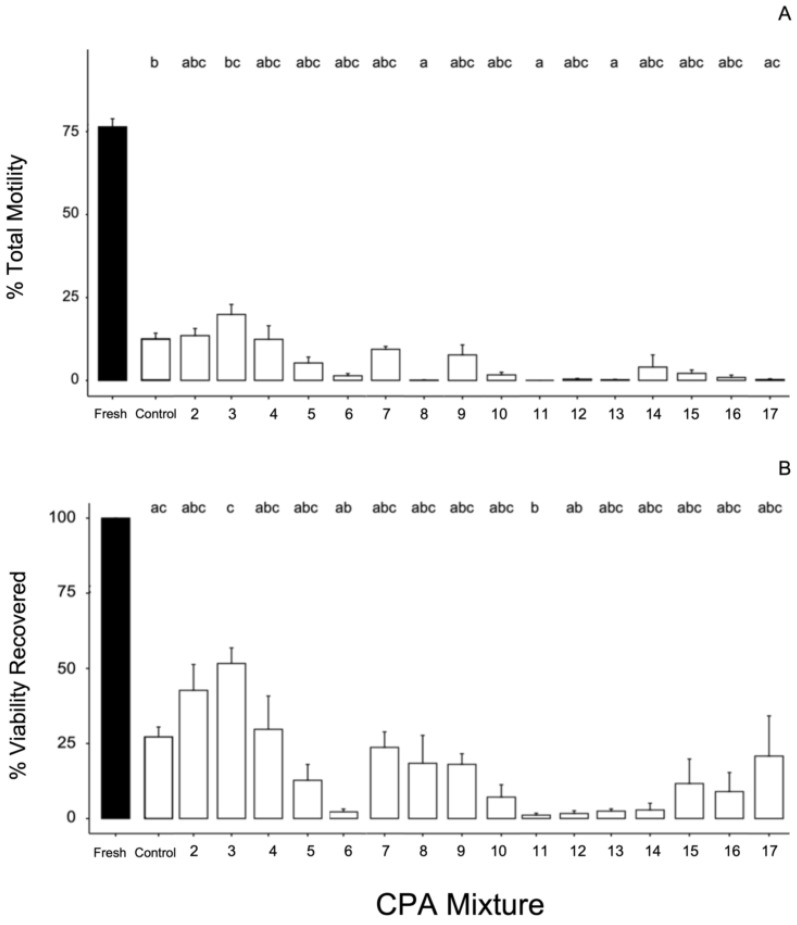
Barplot (**A**): percent total motility of post-thaw sperm of the Louisiana pinesnake cryopreserved using 17 different CPA treatment mixtures; (**B**) percent viability of thawed sperm recovered from fresh sperm. Full CPA treatment recipes are reported in Table 1 as final concentrations following a 1:1 dilution (*v*/*v*) with semen. Four randomized groupings of males were used to test the CPA control mixture (control) and a set of CPA treatment mixtures (Group A = CPAs #3, 4, 5, 7; Group B = CPAs #2, 14, 15, 16; Group C = CPAs #6, 11, 12, 13; Group D = CPAs #8, 9, 10, 17). We tested CPA mixtures based on three categories of experimental tests: CPA base in TEST = CPA #2; Extender base plus additives = CPAs #3–8; and glycerol tests (high vs low) = CPAs #9–17). The bars represent the means and standard deviation for each experimental group. Kruskal–Wallis tests for significance followed by Dunn’s post-hoc test with the Benjamini–Hochberg correction for multiple comparisons to compare CPA treatments; alpha was set at 0.05. The letters above each bar represent significantly different groups, and the black filled bars represent fresh samples and were not included in the statistical tests.

**Figure 2 animals-12-01824-f002:**
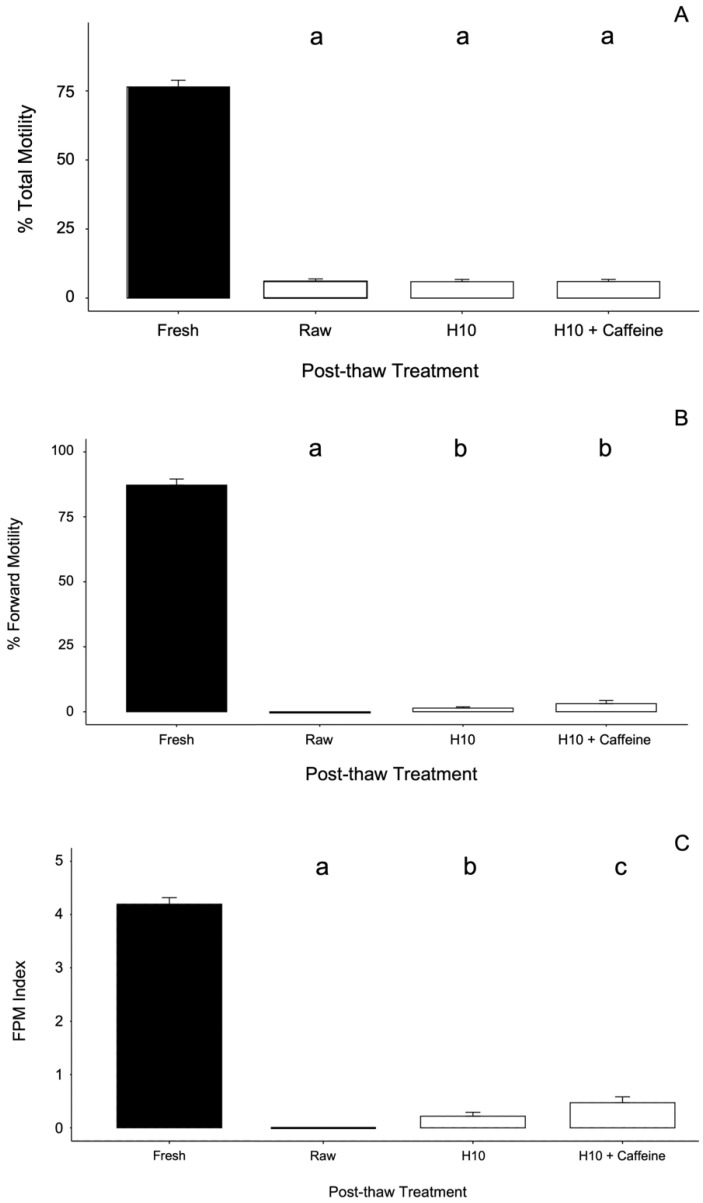
Barplot of (**A**) percent total motility, (**B**) percent forward motility, (**C**) FPM index of thawed sperm of the Louisiana pinesnake: not diluted (Raw), diluted in H10 (H10), or diluted in H10 with 10 mM caffeine (H10 + Caffeine). Bars represent the means and standard deviation for each experimental group. Kruskal–Wallis tests for significance followed by Dunn’s post-hoc test with the Benjamini–Hochberg correction for multiple comparisons to compare CPA treatments; alpha was set at 0.05. The letters represent significantly different groups, and the black filled bars represent fresh samples and were not included in the statistical tests.

**Table 1 animals-12-01824-t001:** Summary table of the 17 CPA treatment mixtures used to test the cryosurvival of sperm collected from the Louisiana pinesnake, *P. ruthveni*. Treatment recipes were reported as final concentrations following a 1:1 dilution (*v*/*v*) with semen. The groups were randomized groupings of males used to test a set of CPA treatment mixtures. Category refers to the three categories of experimental tests: CPA base in TEST, Extender base plus additives, and glycerol tests (high vs. low).

CPA ID	Male Group	CPA Mixture Recipe (Final Conc. *v*/*v*)	Category	N
CPA control	All	Lake’s + 10% DMF + 2% Methanol + 5% Egg	Control Extender base	16
2	B	TEST + 10% DMF + 2% Methanol	CPA base in TEST	4
3	A	Lake’s + 10% DMF + 2% Methanol + 5% Egg + 5 mg/mL BSA	Extender Base + Additives	4
4	A	Lake’s + 10% DMF + 2% Methanol + 5% Egg + 10% FBS	Extender Base + Additives	4
5	A	Lake’s + 10% DMF + 2% Methanol + 5% Egg + 0.01 mM Glycine + 0.009 g/mL Fructose	Extender Base + Additives	4
6	C	Lake’s + 10% DMF + 2% Methanol + 5% Egg + 0.1 M Trehalose	Extender Base + Additives	5
7	A	Lake’s + 10% DMF + 2% Methanol + 5% Egg + 32.37 mM Na-Lactate + 0.50 mM Na-Pyruvate	Extender Base + Additives	4
8	D	Lake’s + 10% DMF + 2% Methanol + 5% Egg + 0.01 mM Glycine + 0.009 g/mL Fructose + 32.37 mM Na-Lactate + 0.50 mM Na-Pyruvate	Extender Base + Additives	5
9	D	Lake’s + 8% DMF + 2% Methanol + 2% DMA + 1% Glycerol + 5% Egg	Low glycerol addition	5
10	D	Lake’s + 8% DMF + 2% Methanol + 1% Glycerol + 5% Egg + 32.37 mM Na-Lactate + 0.50 mM Na-Pyruvate	Low glycerol addition	5
11	C	Lake’s + 6% DMF + 2% Methanol + 1% Glycerol + 5% Egg + 32.37 mM Na-Lactate + 0.50 mM Na-Pyruvate + 0.1M Trehalose	Low glycerol addition	5
12	C	Lake’s + 6% DMF + 2% Methanol + 1% Glycerol + 5% Egg + 32.37 mM Na-Lactate + 0.50 mM Na-Pyruvate	Low glycerol addition	5
13	C	Lake’s + 6% DMF + 6% Glycerol + 5% Egg	High glycerol addition	5
14	B	Lake’s + 16% Glycerol + 5% Egg + 32.27 mM Na-Lactate + 0.50 mM Na-Pyruvate	High glycerol addition	4
15	B	Lake’s + 16% Glycerol + 20% Egg	High glycerol addition	4
16	B	TEST+ 16% Glycerol	High glycerol addition	4
17	D	TEST + 8% Glycerol	High glycerol addition	5

Abbreviations: BSA = bovine serum albumin; DMA = *N*,*N*-dimethylacetamide; DMF = *N*,*N*-dimethyl formamide; DMSO = dimethyl sulfoxide; Egg = clarified egg yolk; FBS = fetal bovine serum.

**Table 2 animals-12-01824-t002:** Summary table of mean values (±s.d.) of fresh sperm concentration, motility, and viability measures of fresh semen samples placed in each of the four experimental CPA groups (A, B, C, D). Below mean values are the results of Kruskal–Wallis test of differences for each semen metric included in the table. M = % motile sperm; MF = % forward-moving sperm; % motile MF = proportion of motile forward-moving sperm; FPM = forward progressive motility.

Male Group	N	Sperm Conc. 10^6^ × mL	% M	% MF	% Motile MF	FPM	% Total Motility	% Viability
A	4	744.3 ± 267.5	10 ± 3.7	61 ± 3.3	86.7 ± 4.8	4.3 ± 0.3	71 ± 4.0	67 ± 18.7
B	4	443.3 ± 330.5	15 ± 10.5	58 ± 9.7	80 ± 12.9	3.6 ± 0.5	72 ± 7.3	56 ± 15
C	5	1241.0 ± 1115.2	4 ± 1.3	76 ± 8.9	95 ± 1.1	4.2 ± 0.4	79 ± 10.2	74 ± 19.1
D	5	1304.8 ± 863.6	11 ± 7.5	70 ± 19.8	85 ± 11.7	4.6 ± 0.4	81 ± 14.2	88 ± 6.4
	*H* value	3.5	7.3	6	7.4	7.6	3.7	8.7
	d.f.	3	3	3	3	3	3	3
	*p* value	0.317	0.062	0.11	0.061	0.055	0.3	0.034 *

* statistically significant at the alpha level of 0.05.

## Data Availability

All associated data are publicly available from the authors or can be accessed at the Figshare archive https://doi.org/10.6084/m9.figshare.20323449.v1.

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
