# Peer review of "Cryopreservation of Sperm from an Endangered Snake with Tests of Post-Thaw Incubation in Caffeine"

_animals, 2022, doi:10.3390/ani12141824_

Round 1

Reviewer 1 Report

This scientific manuscript gives a positive contribution of semen cryopreservation in  Snakes and the  effect of caffeine in  semen quality of thawed semen.

This manuscript is not easy to understand and have many faults.  I have read 3 times (carefully). this manuscript. The principal question is the experimental design .  Also  the obtained  results  are confuse, incomplete and it is not possible to draw reliable conclusions. Now  i will go to some details.  In the experimental design, namely extender formulation for sperm cryopreservation i am totally confused because  all the logical combinations were not performed. We have several extenders, the treatments are confused because we change 2-3 elements in each extender, compare different CPA, using different concentrations, using different snakes or animals, different dates of semen collection. Everything , varies. We cannot identify who are the factors or which CPA,  are accountable  for the better results.   We use, Lake´s, DMF, Test, Glycerol, Methanol, egg yolk, aminoacids, etc.   I suggest a table or pictures with all the possible combinations which are possible,  and the author must say what they compare or which is or are the factors that affect semen quality. I don´t understand the use of H10, or H10 + caffeine. In what frozen semen they are used , i mean which  extender was used  to freeze semen before using Post Thaw additives. I suggest adequate explanation in the text, pictures with experimental design, which criterions were used for extender formulation ;  Only if P<0.05, result are different.  So, some sentences must me changed. Results are very very low. I think the best extender (% total motility) is 2A, which is less than 25 % (Fig 1) (line 317-318) ; But in parts of the text  i see A2.    Why semen was thawed only 31 day after freezing. The number of snakes were 16 or 18?. With so little volumes (ejaculates), and how many replications were made in this experiment (for each extender) ?. I think that temperature variation was very fast (180-182). Define in text what means "were not diluted raw (see fig 2).    Explain lines  321-322, 328-333, 337-339. In lines 339-341, justify the text. Conclusions were not reliable and based in our results. Why AI is recommended?. (339-340). Line  338" was relatively low; I suggest was very low. So based on my comments i suggest a major revision. Some parts are almost   incomprehensible as i have written.

Reviewer 2 Report

Cryopreservation of sperm from an endangered snake with tests of post-thaw incubation in caffeine.

 This manuscript investigates a wide range of semen extenders, CPAs, and post-thaw dilution techniques on semen collected from live males of an endangered snake species, the Pituophis ruthveni.

The study is relevant because it tests the performance of 17 pre-freeze cryoprotective agent (CPAs) treatments and investigates two post-thaw incubation dilution treatments with and without caffeine to optimize sperm cryopreservation. The topic is original because the authors found recovery of membrane viability and post-thaw motility of sperm from the Louisiana pinesnake to be greatly affected by the CPA mixture used during cryopreservation. No other cryopreservation studies are known to have used this combination of CPA mixture to cryopreserve sperm of snakes.

 The results support the conclusions obtained in this study by showing that 12% final concentration of CPA, which included a mix of two CPAs (10% DMFA plus 2% methanol), and BSA to perform the best overall.

Although the cryopreservation of sperm from the Louisiana pinesnake continues to present a considerable challenge, the post-thaw dilution and the addition of BSA to CPA mixtures have improved current methods of cryopreservation for this endangered species.

Reviewer 3 Report

Overall this paper should be published. Could you include descriptive data about ejaculates of this species this would strengthen the paper. There are alot of factors that are at play and not controlled in this study.

Line 40 : >11,000 species

Line 42: Perry et al 2022 has the most comprehensive review of all the reptile cryo literature to date.

Line 47-49: Shouldn't one species be focused on to evaluate reptiles in depth then make minor adjustments following a rich understanding. Each species might be different but methods in evaluation and characterization should be standardized to hasten development.

Line 111: What was the hypothesis? This should be stated.

Line 118-130: Please comment or address the cryptosporidium status of the animals that were collected and the overall health. This population has been impacted significantly by this disease. This type of chronic infectious agent could impact sperm production. This should then be addressed in the discussion.  

Line 174: Did you cryopreserve at a certain spermatozoa concentration as concentration impacts cryopreservation in other species?

Line 175-185- How did you know the addition of CPA itself did not impact motility prior to the freeze? What was happening to the sperm prior to straw loading?  The way the first objective is worded gives the impression you look prior to freezing.

Line 240: you do not talk about correlational analysis in the data analysis. Why is this discussed here?

Figure #1: What viability is this addressing? Is this plasma membrane integrity?

Line 310: This wasn’t a tested hypothesis in this paper. This is speculative.

Line 315: do you use Lake’s buffer as a CPA or as a diluent. Some of the wording earlier makes it appear you are calling it a CPA when it is just a diluent. While on the other hand you can call Test yolk buffer as a CPA as the protiens in the yolk stabilize the PM of the spermatozoa.

Line 325-327- I would remove this contraceptive sentence. There are not well documented contraceptives snakes. This comment of glycerol having a contraceptive impact despite discussion in other species is speculative.
